# Viral Small Terminase: A Divergent Structural Framework for a Conserved Biological Function

**DOI:** 10.3390/v14102215

**Published:** 2022-10-08

**Authors:** Ravi K. Lokareddy, Chun-Feng David Hou, Fenglin Li, Ruoyu Yang, Gino Cingolani

**Affiliations:** Department of Biochemistry and Molecular Biology, Thomas Jefferson University, Philadelphia, PA 19107, USA

**Keywords:** viral genome packaging, terminase subunits, TerS, cryo-EM, X-ray crystallography, bacteriophages, herpesviruses

## Abstract

The genome packaging motor of bacteriophages and herpesviruses is built by two terminase subunits, known as large (TerL) and small (TerS), both essential for viral genome packaging. TerL structure, composition, and assembly to an empty capsid, as well as the mechanisms of ATP-dependent DNA packaging, have been studied in depth, shedding light on the chemo-mechanical coupling between ATP hydrolysis and DNA translocation. Instead, significantly less is known about the small terminase subunit, TerS, which is dispensable or even inhibitory in vitro, but essential in vivo. By taking advantage of the recent revolution in cryo-electron microscopy (cryo-EM) and building upon a wealth of crystallographic structures of phage TerSs, in this review, we take an inventory of known TerSs studied to date. Our analysis suggests that TerS evolved and diversified into a flexible molecular framework that can conserve biological function with minimal sequence and quaternary structure conservation to fit different packaging strategies and environmental conditions.

## 1. Principles of Viral Genome Packaging and the Small Terminase Conundrum

Viral packaging motors are functionally conserved throughout the virosphere, from tailed bacteriophages, nature’s most abundant viruses, to herpesviruses, which represent some of humans’ most common pathogens. These powerful molecular machines generate forces as high as ~60 pN (i.e., ~20–25 times that of myosin ATPase), capable of packaging genomes inside precursor capsids (procapsids or proheads) at speeds approaching 2000 bp/sec [1,2]. The principles governing the packaging of viral genomes are well understood and have been reviewed in detail [3,4,5,6,7,8]. The genome packaging motor consists of three components: a capsid-embedded portal protein, which interrupts the icosahedral capsid symmetry generating a channel for DNA entry inside procapsid [9,10,11]; a large terminase (TerL) subunit that bears all enzymatic activities essential for packaging (i.e., nuclease and ATPase activities); and a small terminase (TerS) subunit, that functions as a DNA recognition factor responsible for binding to packaging initiation sites and handing viral DNA to the packaging motor TerL. In addition, host factors such as small nuclease-associated proteins (HNH proteins) [12] or host integration factors (HIFs) [13] can facilitate the packaging reaction by interacting with TerL and DNA.

Genome packaging is a multi-step reaction, highly diversified in the virosphere, that uses both the virus and host reactants. Although all packaging components have been purified and characterized, the enzymology of packaging is exceedingly complex and challenging to study [14]. In general terms, a packaging reaction can be rationalized into three steps, characterized by different complexes of TerL and TerS. (*i*) Initiation: when TerS and TerL form a ‘pre-packaging initiation complex’ that recognizes packaging initiation sites on viral DNA and introduces nicks into the double-stranded DNA (dsDNA) around the packaging site region. In P22 [15,16], λ [17,18], and HSV-1 [19], the pre-packaging complex can be isolated from infected cells or assembled in vitro from purified components, while in other viruses, the complex is usually transient. (*ii*) Translocation: triggered by TerL docking to the procapsid conformation of the portal vertex to form a ‘packaging terminase motor’ [20,21,22,23]. This machine catalyzes ATP-dependent pumping of one unit length of viral DNA inside procapsid that expands to form a mature virion. TerS stimulates the ATPase activity of TerL through an unknown mechanism. (*iii*) Termination: characterized by the TerL cleavage of the newly packaged DNA and the concomitant binding of tail factors to the portal protein [24], which together release the TerL:DNA complex to initiate a new packaging reaction.

During all three steps described above, TerL undergoes major conformational rearrangements. It is typically monomeric in solution [16,25,26,27,28] but oligomerizes into a pentamer bound to the portal dodecamer [20,22], generating a symmetry mismatch with the portal vertex [29,30]. A high-resolution localized reconstruction of phi29 TerL bound to an immature phi29 capsid [31] found that the TerL oligomer adopts a helical conformation lacking rotational symmetry. Similarly, a recent single-molecule study [32] found that T4 TerL is a flexible pentamer containing one or more inactive subunits. During active packaging, conformational changes in the TerL tertiary and quaternary structures, consistent with an inchworm mechanism [20] or a cyclic–helical symmetry transition [31,33], facilitate the chemo-mechanical coupling of ATP hydrolysis to DNA translocation. However, in most of the reaction steps described above, TerS is not part of the packaging motor, or at least, in vitro, does not appear to play a direct role in genome packaging. For instance, bulk genome packaging systems developed for phages T4 [34], T3 [35], λ [36], and P22 [15,37,38] did not require TerS in vitro. Similarly, TerS inhibited packaging in a defined in vitro packaging system carried out in the presence of purified T4 [39,40,41] or SPP1 [42] components. Along the same lines, a single-molecule packaging assay for T4 was strongly inhibited by an excess of TerS [2,43] and, similarly, the overexpression of TerS in a complementation assay reduces phage E217 infectivity [28]. Thus, TerS is an essential viral subunit whose function is not easily recapitulated in vitro, suggesting a strictly concentration-dependent function and short kinetic window of action.

## 2. Conservation of TerS in the Virosphere

Genetic evidence of the existence of a gene product involved in recognizing packaging initiation sites dates to 1982 [44]. This gene product was first isolated in phage λ (gpNu1) [45,46] and then found in the *Salmonella* phage P22 [47] and *Bacillus subtilis* phages SPP1 and SF6 [48]. Since then, TerS has been identified in many other phages, including thermophilic phages, typically neighboring TerL, and is arranged in a small operon. However, the physical proximity of the TerS and TerL genes is not a universal feature. Lokareddy et al. found the TerS gene of phage E217 is ~58 kbs away from the TerL gene, on the opposite side of the genome [28].

We performed phylogenetic and amino acid conservation analyses of TerS from 20 phages and eight herpesviruses. Analyzing the phylogenetic tree of TerS in phages revealed tree branches with limited confidence [49], suggesting poor evolutionary conservation, except for very similar, and in some cases almost identical, phages infecting the same bacterium (e.g., SF6/SPP1 for *Bacillus subtilis*; G20c/P74-26 for *Thermus thermophilus*; NV1/PaP3 for *Pseudomonas aeruginosa*; and T3/T7, P21/λ, and T4/44RR for *E. coli*). TerSs from phages infecting different bacteria occupy standalone groups (Figure 1a), with less than ~15% amino acid identity on average (Figure 1b). For instance, Sf6 and P22, both P22-like phages [50], have highly divergent TerS that cluster away from each other, with the former being more related to the *Myoviridae* SK1 (identity/similarity = 14%/27%) than the *Podoviridae* P22 (identity/similarity = 12%/25%). Even TerS from two common *Escherichia*-phages, such as T7 and T4, have significantly diversified, with sequence identity and similarity scores of only 12% and 19%, respectively. By comparing TerS from two *Myoviridae*, he *Actinomycetes* phage SK1 and E. coli phage T4, the sequence identity/similarity is just 8% and 16%, respectively, dropping to only 5/8% when compared to the Podoviridae T7 (Figure 1b). Thus, TerS has diverged enormously in bacteriophages regardless of the tail morphology or packaging strategy (discussed in Section 5). As pointed out by Casjens and Thuman-Commike, TerS sequence diversity is more significant than TerL or portal protein in tailed bacteriophages [50], making it difficult to annotate the TerS gene in new phage genomes.

TerS is an entirely different protein in herpesviruses, unrelated to phages TerS, and encoded by a gene about four times the size of phage TerS with no detectable sequence similarity [51,52]. Unlike phages, TerS is more conserved in herpesviruses (Figure 1a). A phylogenetic tree analysis of eight TerS sequences similar to human herpesvirus 1 (HHV-1, also known as herpes simplex virus 1, or HSV-1) showed that pUL28 readily sorted TerS into three subfamilies, corresponding to alpha, beta, and gamma herpesviruses with a sequence identity/similarity of 16–20% and 33–34%, respectively, and similar amino acid length (Figure 1b). In summary, TerS has diversified dramatically in bacteriophages, but it is more conserved in herpesviruses.

**Figure 1 viruses-14-02215-f001:**
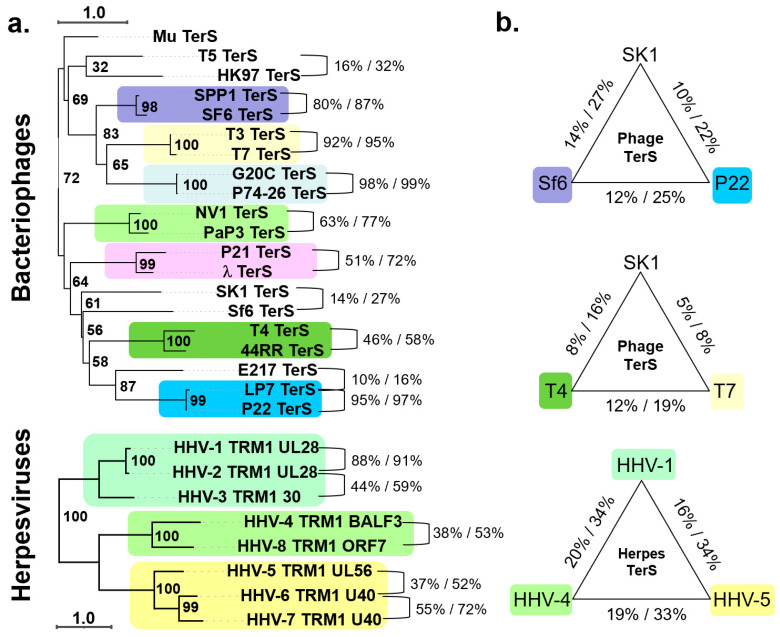
TerS conservation in bacteriophages and herpesviruses. (**a**) Phylogenetic trees of 20 representative phage TerSs (top panel) and 8 pUL28-homologs (bottom panel). The scale bar represents the estimated evolutionary distance. The supporting numbers in the tree refer to the percentage of bootstrap distribution based on 1000 resamples. TerSs that share significant similarities are grouped and colored consistently. Seven groups of TerSs were identified among the 20 phages analyzed here, and only three for herpesviruses. The sequences were aligned with Clustal Omega [53] and converted to FASTA/Phylip format using EMBOSS Seqret [54]. The phylogenetic trees were analyzed in PhyML3.0 [55] and drawn using iTOL [56], which re-roots at the mid-point. (**b**) Sequence identity and similarity between representative TerSs from the groups in panel (**a**) were calculated using SMS [57].

## 3. Diversification of the TerS Fold and Oligomeric State in Bacteriophages

The structure of several TerSs from bacteriophage and HSV-1 has been investigated using biophysical methods, providing a useful framework for understanding their assembly and activity. Three-dimensional structures have been obtained for TerS from bacteriophages that infect *Escherichia coli* (i.e., λ [58], T4-like 44RR [59], HK97 [60]), *Shigella flexneri* (i.e., Sf6 [61,62]), *Salmonella enterica* (i.e., P22 [63,64,65,66]), *Bacillus subtilis* (i.e., SF6, SPP1 [67]), *Pseudomonas aeruginosa* (i.e., PaP3 [68], NV1 [68], E217 [28]), and *Thermus thermophilus* (i.e., G20c [69], P74–26 [70]) (Table 1). All atomic structures of phage TerSs reported between 2002 and 2019 were solved using X-ray methods or used NMR for a fragment of phage λ TerS [58]. Since the advent of the cryo-EM revolution [71], single-particle reconstructions were reported for phage P74-26 [70] and E217 [28] TerS, as well as HSV-1 pUL28 (TerS), determined in complex with pUL15 (TerL) and pUL33 [72]. As pointed out above, there is no structural similarity or evolutionary conservation between phages and herpesviruses TerSs, which are different proteins of similar function (described in Section 7). Here, we will focus on analyzing structural features conserved in bacteriophages.

All phage TerSs form ring-like oligomers of different stoichiometry, predominantly nonamers [63,64,65,67,68,70,73], although octamers were observed for the *Podoviridae* Sf6 [62], decamers for the *Myoviridae* E217 [28], and 44RR TerS [59] crystallized as a mixture of undecamer and dodecamers (Figure 2a). The oligomeric structure generates a fully hydrated internal channel that varies significantly in diameter, between 9 and 52 Å (Table 1), i.e., large enough to accommodate hydrated dsDNA in most cases, with some exceptions such as TerS from Sf6, PaP3, HK97, and SF6.

The TerS protomer is conserved in size, between ~15 and 21 kDa (Table 1), and is built by three structurally conserved regions and one variable moiety (Figure 2b). The conserved regions include an N-terminal “helix-turn-helix” (HTH) motif making up most of the putative DNA-binding domain; a central “oligomerization helical core” consisting of a helical hairpin that delineates a channel lumen; and a “stalk” that extends the central oligomerization core and can be formed by α-helices or β-strands. The fourth and most variable region is a flexible “C-terminal moiety“ that extends the stalk. Overall, the four TerS regions have low structural and amino acid sequence conservation.

The N-terminal HTH is the most conserved structural feature, shared by all TerSs and involved in DNA binding (see Section 6). This domain was not observed in the crystal structure of 44RR TerS [59] but can be predicted in the amino acid sequence. In SF6 [67,74] and PaP3 [68], HTHs are connected to the oligomerization core by flexible, protease-susceptible linkers, whereas TerS from the thermophilic phage P76-26 [70] has HTHs rigidly held together. Unlike the HTH, the oligomerization core varies for the length of the two α-helices and the presence of an inserted β-hairpin that generates a chapel-like cap structure at one end of the channel [63,67]. Differences in the oligomerization helical core (Figure 2b) are likely responsible for the oligomerization stoichiometry observed in different phages (Table 1) and the polymorphic assembly observed in vitro [59], as well as the internal channel diameter, which varies widely (Table 1). TerS channel diameter and quaternary structure stoichiometry (Table 1) do not appear to be pre-requisite for function.

The stalk is also highly divergent. It extends the helical channel, forming a β-barrel in TerS from Sf6 [62], P22 [63], SF6 [67], and PaP3 [68], but is missing in TerSs from phages P74-26 [70], HK97 [60], and T4 [59], where the oligomerization core helices are longer. Finally, C-terminal of the stalk is a flexible moiety typically not seen experimentally in the crystal structures of TerSs. In P22, a subset of TerS particles analyzed by negative-stain electron microscopy revealed an elongated tail, possibly formed by a folded conformation of the C-terminal moiety [63]. Similarly, the C-terminal moiety of E217 TerS contains a putative α-helix, hypothesized to insert into the channel of another TerS, promoting the formation of high-order oligomers [28]. Overall, the repertoire of TerS structures solved thus far suggests that the function of this viral protein can be conserved using different quaternary structure assemblies. We speculate that the TerS protomer harbors sufficient structural determinants for biological activity, and the oligomer amplifies them, possibly by avidity effects. Thus, TerS channel diameter and quaternary structure stoichiometry (Table 1) have diversified greatly in tailed bacteriophages to the point that a conserved tertiary and quaternary structure do not appear to be pre-requisites for function.

## 4. TerS association with TerL and Modulation of TerL Catalytic Activities

TerL and TerS assemble during genome packaging; however, with a few exceptions, a biochemically stable TerS:TerL complex has proven challenging to form in vitro as terminase subunits usually interact transiently. The TerS:TerL complex is stably populated only in three viruses studied thus far. Phage λ terminase subunits form a stable ~114.2 kDa heterodimer, consisting of one TerL (gpA) bound to two TerS subunits (TerL_1_:TerS_2_), in equilibrium with a 13.3 S species of ~530 kDa, consisting of four protomers [17,18]. The C-terminal region of phage λ TerS binds TerL [75,76], as also reported for P22 TerS [63], where the binding site in TerL was mapped to the first 23 N-terminal residues [16]. The phage λ heterotrimeric TerL_1_:TerS_2_ complex is devoid of catalytic activity [17,18,77] and likely represents a pre-packaging assembly poised for activation. Similarly, a TerL:TerS stable complex was purified from cells infected by the *Salmonella*-phage P22 [15]. Recombinant P22 TerS and TerL subunits also assemble into a complex in vitro [63] or when co-expressed in bacteria [16]. A low-resolution reconstruction of the P22 TerS:TerL holoenzyme revealed a 9:2 stoichiometry [16], equivalent to one TerS oligomer bound to two TerL, although the specimen used for this reconstruction was heterogenous and contained other assemblies with more than two TerL subunits. Mapping studies revealed that the N-terminus of P22 TerL (residues 1–58) contains a minimal TerS-binding domain sufficient for association with TerS C-terminal residues 140–162 in vitro [16]. Finally, in herpesviruses, TerL and TerS form a stable complex in infected cells with a third regulatory subunit, pUL33 in HSV1, which may function as a chaperone or assembly factor [19,78]. The estimated mass of this complex was consistent with a ratio of 1:1:1, suggesting a pre-packaging conformation. However, a recent cryo-EM reconstruction of the HSV-1 terminase complex composed of TerL (pUL15), TerS (pUL28), and pUL33 [79] revealed a hexameric assembly of this trimeric terminase complex, hypothesized to be the packaging conformation of the motor.

In all other bacterial viruses, the association between TerS and TerL is transient, difficult to capture in vitro using purified components, but relatively easy to probe in vitro. TerS strongly modulates both TerL catalytic activities; it can stimulate the ATPase activity of TerL [63,80,81,82] while repressing [59,73,83,84,85] or activating [72] TerL nuclease activity. For instance, T4 TerS gp16 can stimulate TerL weak intrinsic ATPase activity 50 to 100-fold in vitro [81]. This TerS regulatory function is likely mediated by the N- and C-terminal domains and is thought to prevent TerL’s otherwise potentially suicidal activity by restricting its activity and avoiding random cutting in the virus genome [39]. In P22, the TerS-mediated stimulation of TerL ATPase activity depends on *pac* DNA, suggesting the ATPase activity is stimulated only when TerS, TerL, and DNA come together [63]. This mechanism could avoid activating the packaging motor when the incorrect DNA sequence is presented to TerL. In addition, phage λ TerS binds nucleotides (ATP, ADP, GTP, and GDP) with low affinity [79], and nucleotide binding regulates TerS DNA-binding interactions. 

The TerS stimulation mechanisms of TerL ATPase activity are poorly understood. It was suggested that TerS stimulation of the TerL ATPase activity occurs by a mechanism reminiscent of GTPase-activating proteins such as RCC1, which binds and stimulates the nucleotide hydrolysis of the GTPase, ran by the physical stabilization of the P-loop instead of providing an arginine finger in trans [86]. In analogy, TerS interaction with TerL could aid in positioning a TerL arginine finger into the catalytic pocket to enhance ATP hydrolysis [86]. These predictions remain speculative without a high-resolution structure of the TerS:TerL complex. Overall, TerS inhibitory function in defined packaging systems [2,39,40,41,43] and its regulatory role of TerL ATPase and nuclease activities suggest that TerS is unlikely bound to TerL during active packaging, but the TerS:TerL complex forms in preparation to packaging, possibly before or during binding to packaging initiation sites.

## 5. TerS Function in *cos* versus *pac* Packagers

The small terminase primary function is to recognize viral DNA and bring it to TerL, but the way TerS recognizes DNA varies depending on a virus’ packaging strategy. Provided that the substrate for genome packaging is a concatemer DNA molecule in all phages and herpesviruses, but only a single genome unit is inserted inside a procapsid (with some additional host DNA in transducing phages), a major mechanistic difference exists between phages that use a *cos* versus a *pac* sequence [87].

In *cos* packagers such as phage λ [88], the *cos* sequence serves as both the packaging initiation site used for genome recognition during genome packaging and a specific packaging termination sequence. The *cos* site represents a point of junction between two genome units within a concatemer that results from annealing two cohesive ends at the *cos* sequence. For instance, phage λ carries 12 base-long single-strand extensions (5′-GGGCGGCGACCT-3′) surrounding its chromosome that generate a *cos* site upon entry into a host cell. *Cos* sites are recognized by TerS and processed by TerL. Specifically, TerL introduces precisely staggered nicks in the *cos* sequence, while TerS binds the *cos* site in a sequence-specific manner, with detectable biochemical affinity in vitro, as seen for phages λ (gpNu1) [89] and PaP3 TerS [68]. As a result of the TerS:*cos* site specificity, *cos* packagers can accurately package one genome unit at a time without terminal duplications. This specificity is also observed in vitro, where TerS from *cos* packagers [45,58,90] binds specifically to their cognate *cos* sequence. For instance, recombinant PaP3 TerS binds a 20mer *cos* dsDNA oligonucleotide (5′-GCCGGCCCCTTTCCGCGTTA-3′) with an equilibrium binding constant Kd ~10 μM [68].

In contrast, viruses that use the head-full packaging mechanism, such as P22 [91], Sf6 [92], SPP1 [93], and SF6 [67], have a *pac* sequence, which is the putative recognition site for TerS. The packaging reaction is initiated by a first cut in the proximity of a *pac* site that consists of a 22-bp asymmetric sequence in the TerS gene for P22 [92] or multiple points of contact flanking the site where TerL makes an initial cut in SPP1 [94]. Genome packaging proceeds possessively in *pac* packagers and is terminated by a non-specific cut when the procapsid is full (hence the name ‘head-full’ packaging). The termination cut is also the start of the packaging for the next chromosome along the concatemer. Unlike P22, in SPP1, the *pac* site is estimated to be used only once every four packaging events [95]. Notably, TerS association with a *pac* sequence, located within the TerS gene in P22, is supported mainly by genetic evidence [92], but attempts to measure TerS association with a dsDNA oligonucleotide containing the P22 *pac* sequence (5′-AGAGAAGATTTATCTGAAGTCG-3′) were unsuccessful [63,66]. Similarly, the interaction of TerS with *pac* site in other *pac* packagers such as Sf6 [61,62], P76-26 [70], SF6 [67], and SPP1 [96] was weak and difficult to study quantitatively in vitro, mainly resulting in sequence-independent bandshift. An exception to this rule came from a report that the SPP1 TerS binds DNA with nanomolar affinity, inducing significant bending in the double helix [94]. As no negative control was provided in this study, it cannot be ruled out that a contaminating DNA-binding protein was responsible for putative DNA binding (not detected in later studies [96]). Moreover, SF6 TerS [67], which is 71% identical in amino acid sequence to SPP1, also binds DNA weakly and non-specifically in vitro [96].

More difficult to decipher is the association of T4 TerS with DNA. There is no unique *pac* site in the T4 genome, and T4-like phages [97] package their large genomes (up to 250 kbs) [98] by a mechanism whereby no *pac* site is recognized, and packaging initiates randomly. This packaging strategy is successful because T4-phages degrade the host DNA, ensuring that phage DNA is the main substrate available for packaging [98]. The T4-like phage 44RR (56) crystal structure revealed a simple helical hairpin lacking a canonical HTH [46,59]. The oligomeric structure of T4 TerS possibly functions by creating an outer surface for DNA adsorption [97] that is then passed to the pentameric TerL motor. Recombinant T4 TerS gp16 binds DNA very weakly in vitro, showing a preference for longer DNA fragments [99]. Additionally, TerS from the *Myoviridae* phage E217 did not show appreciable DNA binding activity in vitro [28], possibly suggesting a packaging strategy similar to T4.

We reviewed all HTHs from TerS structures deposited in the PDB that were subjected to site-directed mutagenesis to identify residues involved in DNA binding (Figure 3). However, HTHs from different TerS are not easily superimposable, and helices making up the HTH have distinct orientations and lengths. Focusing on TerS from *cos* packagers that bind DNA specifically, in phage λ TerS, residues K5 and K6 (in helix α1) and S15, R17, T18, Q20, N21, and Q23 (in helix α2) are essential for DNA binding [58]. Similarly, in PaP3 TerS HTH, a double Ala-mutant in helix α1 (K17A/K19A), which displayed severely reduced DNA binding, was further enhanced by introducing additional mutations in helix α (K33A) and α3 (R49A/R56A/K57A). Thus, the first HTH α-helix appears to be essential for DNA binding, while additional basic residues strengthen the association with DNA. In contrast, the residues shown to reduce non-specific DNA binding in *pac* packagers (Figure 3) are scattered over the surface of several α-helices. In Sf6 TerS, mutations K33E (helix α2), R48A (helix α3), and K59E (helix α4) showed considerably reduced non-specific DNA binding, alone or in combination. Furthermore, K6E (helix α1) drastically affected TerS DNA binding activity, possibly suggesting that these residues are in close proximity and presumably involved in electrostatic interactions with DNA phosphate backbones [61]. In SF6, viral DNA recognition via HTH was shown to be non-specific, possibly through multiple sequence-independent interactions with an HTH [96].

## 6. Mechanisms of DNA Recognition

A key question still unanswered is how TerS recognizes DNA. Two mainstream models have been proposed to explain how TerS recognizes viral DNA: the ‘threading model’ [63] and the ‘nucleosome model’ [58,59,62,66,67] (Figure 4a,b). Supporting evidence for either model has been scarce and inconclusive.

In the threading model, viral DNA is threaded through the central channel, similar to a helicase or portal protein (Figure 4a). This model was never tested structurally by solving a complex of TerS with DNA, but mutational studies in T4 led to the exclusion of the threading model, at least for this phage. Furthermore, the structures of TerS from phages Sf6 [61,62], SF6 [67], HK97 [60], and PaP3 [68] revealed a channel diameter smaller than 20 Å (Table 1, Figure 2a), which was too narrow to accommodate hydrated dsDNA. This structural evidence makes the threading model for DNA passage unlikely, at least assuming that DNA is threaded inside the channel as a double-stranded polymer [97].

For the nucleosome model, the existence of outward-pointing N-terminal HTHs, a domain involved in DNA binding, prompted many groups to hypothesize that TerS recognizes DNA via its N-terminal domains akin to a nucleosome core particle where DNA wraps around histone proteins (Figure 4b). This model was bolstered by foot-printing data using recombinant SPP1 TerS [94]; however, as pointed out above, this study did not provide a negative control, and similar results were not supported in later studies [96]. Unlike histones that bind tightly to DNA via electrostatic interactions, most TerSs associate with DNA weakly and non-specifically in vitro, making it unlikely that multiple HTHs are simultaneously involved in DNA binding. All models for DNA-binding proposed thus far suffer from major weaknesses, seemingly because neither a nucleosome-like binding nor threading DNA through a channel can explain sequence specificity, a vital function of TerS, especially in *cos* packagers. Not surprisingly, DNA wrapping is a binding mode used mainly by sequence-independent proteins such as histones that lack nucleobase specificity but make electrostatic contact with the DNA phosphate backbone. Similarly, threading the double helix through a protein channel is intrinsically sequence-independent, as seen for portal proteins [100] or DNA ejectosomes [101].

Alternative models have been proposed to reconcile the biochemical data and explain nucleobase-specific recognition. Black et al. proposed an extension of the nucleosome model, referred to as the ‘twin ring *pac* synapsis’ [102] (Figure 4c). According to this model, T4 TerS functions as two stacked rings that recognize two apposed *pac* sites to gauge concatemer maturation adequate for packaging initiation. This model, supported by several lines of evidence, may be limited to T4, as twin packs of TerS have not been observed in other phages.

The ‘thimble model’ was proposed for the *pac* packager P22 (Figure 4d). The TerS recognition of pac-containing DNA sequences in this phage results from DNA-binding motifs in the N-terminal HTHs and the C-terminal tail [66,99]. Deleting the C-terminal tail obliterates both DNA binding and TerL association [63]. According to the ‘thimble-model’, the TerS oligomer serves as a cap to trap DNA, either the end of dsDNA or single-stranded DNA from a nicked dsDNA [63]. This thimble-like structure allows the C-terminal moiety of P22 TerS to fold back and insert into the DNA groove, making sequence-specific contacts. A similar mechanism could be envisioned for TerS N-terminal HTHs, which may be capable of flexibly folding back and reading the DNA nucleobases.

Niazi et al. suggested a conceptually distinct model for the *cos* packager PaP3, where TerS makes strong and stable interactions with the *cos* DNA site (Figure 4e). Specifically, it was proposed that TerS [68] may recognize a *cos* site by lateral interdigitation using adjacent HTH domains. In this model, pairs of HTH motifs use their intrinsic structural plasticity to make asymmetric and sequence-specific contacts with the *cos* site. Similar flexibility in how HTHs connect to the TerS oligomerization core was observed in SF6 TerS [67]. DNA interdigitation could also allow more copies of TerS to simultaneously bind DNA, as observed in high-order complexes of TerS with DNA reported for SPP1 [94], λ [58], and Sf6 [62]. However, lateral interdigitation cannot be universal to all TerS. This model would not apply to the thermophilic phage P74-26 [70], which has HTHs rigidly held together, unlike most TerSs from mesophilic viruses. In conclusion, the exact mechanisms of TerS DNA binding remain elusive and controversial, especially without a structure of TerS bound to DNA.

## 7. Functional Conservation of TerS in Herpesviruses

TerS and TerL are functionally conserved in herpesviruses that also contain a third subunit (i.e., pUL33 in HSV-1 or pUL51 in HHV-5), thought to function as a chaperone or assembly factor [51,52,78]. TerL (i.e., pUL15 in HSV-1 or pUL89 in HHV-5) is a *bona fide*
*ortholog* of phage TerL, which shares significant amino acid homology [4], including an N-terminal ATPase domain with Walker A/B motifs and a C-terminal nuclease domain [103,104] superimposable to that of many phages [73,105] (Figure 5a,b).

In contrast, there is no sequence homology between bacteriophage and herpesvirus TerS, which is evolutionarily and structurally a different protein (Figure 1a,b). The gene encoding herpesvirus TerS (i.e., pUL28 in HSV-1 or pUL56 in HHV-5) is highly conserved among all herpesviruses with sequence identity between 16% (between alpha and gamma herpesviruses) and 88% (within alpha herpesviruses) [106], suggesting an essential and ancient function (Figure 1a,b). Because herpesvirus TerS is larger than TerL, some authors referred to the former as the ‘large’ terminase, generating a great deal of confusion in the literature [107,108,109]. Cryo-EM reconstruction of the HSV-1 terminase complex composed of TerL (pUL15), TerS (pUL28), and pUL33 was recently reported [79] (Figure 5a). The structure revealed an intimate association between TerS, a large (M.W. ~85 kDa) all-α-helical protein, and TerL (M.W. ~81 kDa), which accounts for a total interaction area of ~6700 Å^2^. TerS makes contacts with both TerL active sites, inserting an α-helix (res. 478–507) against the nuclease domain and surrounding the entire N-terminal ATPase domain with its C-terminal core (res. 532–775) (Figure 5a). TerS contains two zinc fingers, one intra-molecular and the other inter-molecular, generated by residues from both pUL28 and pUL33 (Figure 5b). The regulatory subunit pUL33 (M.W. ~13 kDa) interacts with the TerS C-terminal domain and does not directly contact the TerL ATPase domain (Figure 5a,b). Unlike pentameric phage TerL, the pUL28:pUL15:pUL33 complex assembles into a hexameric quaternary structure [72] with an external diameter of ~225 Å and a height of ~100 Å, significantly larger than bacteriophage TerL motors or hexameric ATPases of the conserved ASCE superfamily [110].

A low-resolution negative-stain EM structure of HHV-5 (also known as human cytomegalovirus, or HCMV) TerS (pUL56) [111] revealed two U-shaped TerS protomers sitting next to each other (i.e., pUL56 exists primarily as a dimer [111]). As seen in the homologous HSV-1 pUL28 (Figure 5b), HCMV TerS is a helical solenoid protein [112] similar to importin-β-like factors [113]. Interestingly, herpesvirus terminase subunits contain potent nuclear localization signals (NLSs) used to shuttle between the cytoplasm and nucleus of infected cells, where viral replication occurs [51]. Specifically, HCMV TerS has a functional NLS [109] between residues 814 and 829 (Figure 6a) [108,109]. Co-crystallization studies with human importin α revealed that HCMV TerS NLS intimately binds the major NLS-binding pocket of importin α, making at least six close contacts [106] which are essential to promote pUL56 nuclear import in a complex with the adaptor importin α (Figure 6b) and the receptor importin β. In the nucleus, pUL56 binds *pac* sites on the HCMV genome. The mechanisms of genome packaging are significantly less well-understood in HCMV than in bacteriophages. pUL56 is known to bind concatemeric DNA on *cis*-acting packaging signals termed *pac* motifs [107]. Specifically, *pac**1* and *pac2* lie within regions of HCMV DNA that are coincident with *cis*-acting cleavage elements recognized by TerL pUL89. Consistent with its function as a DNA-binding protein, pUL56 contains a zinc finger between residues 190 and 220 [107,114,115]. As previously noted, two Zn fingers were identified in the cryo-EM reconstruction of HSV-1 pUL28 (Figure 5b). Finally, pUL56 dimerization [111] strengthens the association with the palindromic *pac1* [107] motif on the HCMV genome.

pUL56 TerS is the cellular target of letermovir (M.W. ~572.5 Da), an FDA-approved antiviral agent of great potential in the treatment of HCMV opportunistic infection in transplant patients, currently under development by Merck & Co. Letermovir exhibits outstanding anti-HCMV activity in vitro (EC_50_ = 5.1 ± 1.2 nM) and a selectivity index exceeding of 15,000 [116]. Letermovir targets pUL56 [116,117] and is potent against clinical isolates and in a mouse xenograft model [116]. Importantly, this small molecule is highly effective against virus strains resistant to currently approved antivirals. Single L241P or R369S amino acid substitutions in pUL56 are necessary and sufficient to produce letermovir resistance in cell lines [117]. A sequence alignment comprising 61 UL56 sequences in GenBank, including 41 sequences from clinical isolates, revealed a letermovir resistance region spanning residues 230–370 of pUL56 [118]. These residues likely harbor a high-affinity binding pocket for this letermovir.

## 8. Conclusive Remarks: Challenges and Gaps to Be Filled

Four decades after the identification of a gene product required to recognize *pac* sites [44], the biology of viral TerS is still filled with gaps. TerS is undoubtedly a DNA-binding factor, but the mechanisms of DNA binding continue to be unclear and controversial, and so is the way that TerS is associated with TerL, the coordination, and the space–temporal role played in genome packaging. Reviewing the literature on TerS led us to several general conclusions, listed below:(i)The gene encoding TerS has diverged more and faster than TerL or portal proteins, suggesting that this terminase subunit exerts a function that can be conserved with significant variations on the same structural framework. TerS is functionally conserved in herpesviruses but fundamentally reinvented from bacteriophages.(ii)TerS is essential in vivo but dispensable in vitro where an excess of TerL, viral DNA, and ATP are sufficient to promote energy-dependent genome packaging.(iii)Phage TerS are oligomeric in bacteriophages, but the stoichiometry of oligomerization varies mainly with nonamers, decamers, and octamers. This suggests that TerS can retain its function with a different stoichiometry of oligomerization. In herpesviruses, TerS is larger than TerL and adopts a helical solenoid-like structure.(iv)TerS interacts with TerL to promote genome packaging, but this association can be very transient or remarkably stable, suggesting that the heterotypic association of terminase subunits has diverged significantly, even in closely related phages.(v)TerS association with DNA is fundamentally different in *cos* versus *pac* packagers. *Cos* packagers encode TerSs that make strong and saturable interactions with dsDNA. In contrast, TerS from *pac* packagers makes weak, sequence-independent contacts with dsDNA.(vi)Both threading and nucleosome models are incomplete and unsupported by biochemical data. Neither model can explain the sequence-specific recognition of *cos* sites, a vital function of TerS. Whatever model is contemplated must consider how TerS can read DNA nucleobases. TerS contains Zn-binding domains in herpesvirus, possibly responsible for specific DNA recognition.(vii)HCMV TerS (pUL56) is the target of letermovir, an FDA-approved antiviral agent of great potential in treating HCMV opportunistic infection in transplant patients.

In conclusion, forty years since its discovery, the small terminase subunit remains the most enigmatic of all factors involved in viral genome packaging. Structural studies of TerS in complex with DNA are needed to decipher how the identified DNA-binding determinants in TerS result in the sequence-specific recognition of packaging initiation sites, an essential activity in the virosphere. The hope is that cryo-EM will allow scientists to capture TerS or a TerS:TerL complexes bound to DNA, thus reconciling decades of sparse and often conflicting biochemical observations.

## Figures and Tables

**Figure 2 viruses-14-02215-f002:**
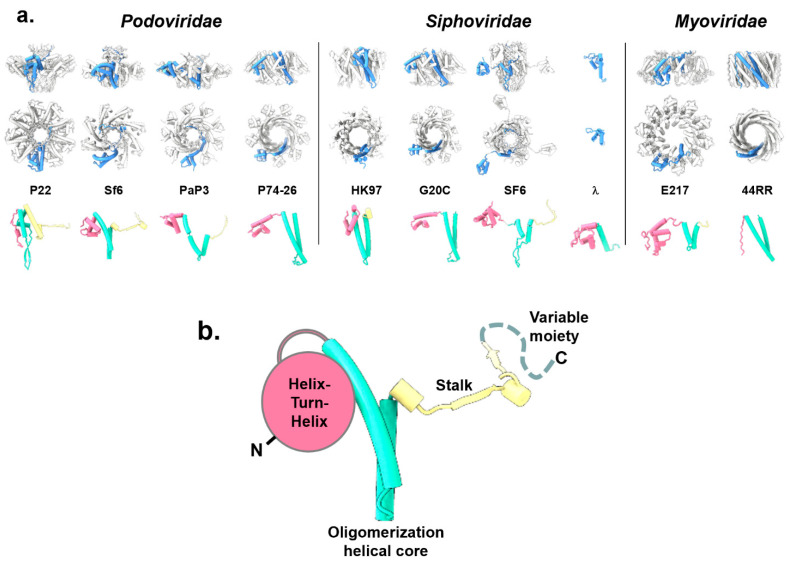
Structures of phage TerSs deposited in the PDB and EMDB. (**a**) Quaternary structures of TerS proteins from *Podoviridae*, *Siphoviridae*, and *Myoviridae* are shown as side and top views. All structures are shown in scale, with the protomer “A” colored in blue and the rest of the oligomer in gray. The tertiary structures of each TerS protomer are shown under the oligomer color-coded to highlight the N-terminal HTH-domain (magenta), the oligomerization core (cyan), and the stalk (yellow). All TerS oligomers and protomers are in scale. (**b**) A schematic diagram of the TerS-fold that consists of three conserved structural regions and a variable C-terminal moiety (dashed line).

**Figure 3 viruses-14-02215-f003:**
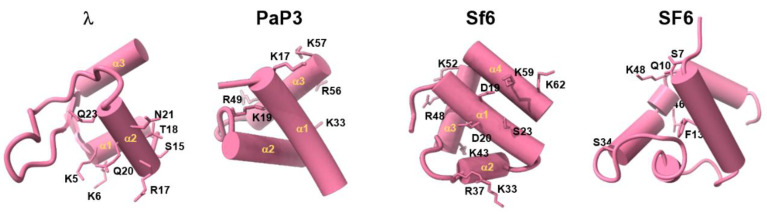
TerS N-terminal helix–turn–helix motif involved in DNA binding. Structures of experimentally determined HTH motifs from phages λ, PaP3, Sf6, and SF6. Residues involved in DNA binding and identified through site-directed mutagenesis are shown as sticks.

**Figure 4 viruses-14-02215-f004:**
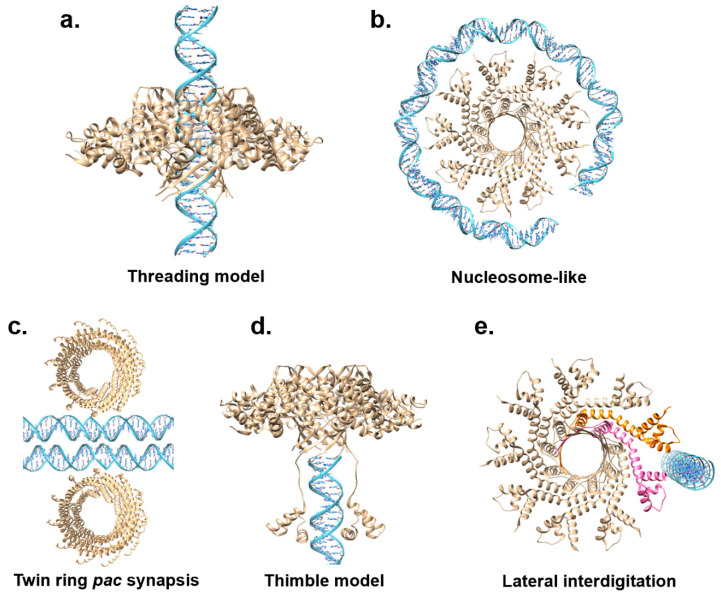
Models for TerS-mediated DNA recognition. (**a**,**b**) Models inconsistent with sequence-specific recognition of nucleobases: the nucleosome model and the threading model. (**c**,**d**) Models that explain sequence-specific recognition of DNA nucleobases. PaP3 TerS was used as the template for panels (**a**,**b**,**d**,**e**), while panel **c** shows phage 44RR TerS.

**Figure 5 viruses-14-02215-f005:**
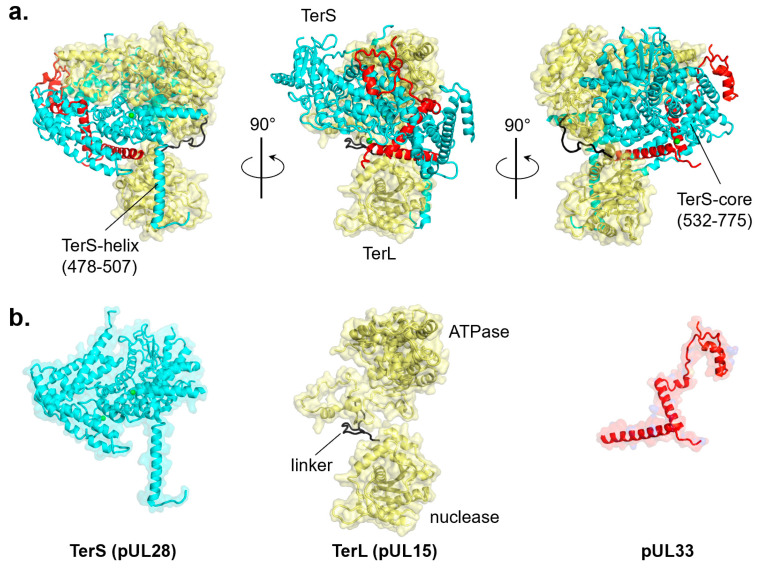
Cryo-EM reconstruction of the tripartite HSV-1 terminase complex (PDB id 6M5S). (**a**) Ribbon diagram of the three terminase subunits: TerS (pUL28), TerL (pUL15), and pUL33, colored in cyan, yellow, and red, respectively. (**b**) Tertiary structures of individual terminase subunits are color-coded as in panel (**a**).

**Figure 6 viruses-14-02215-f006:**
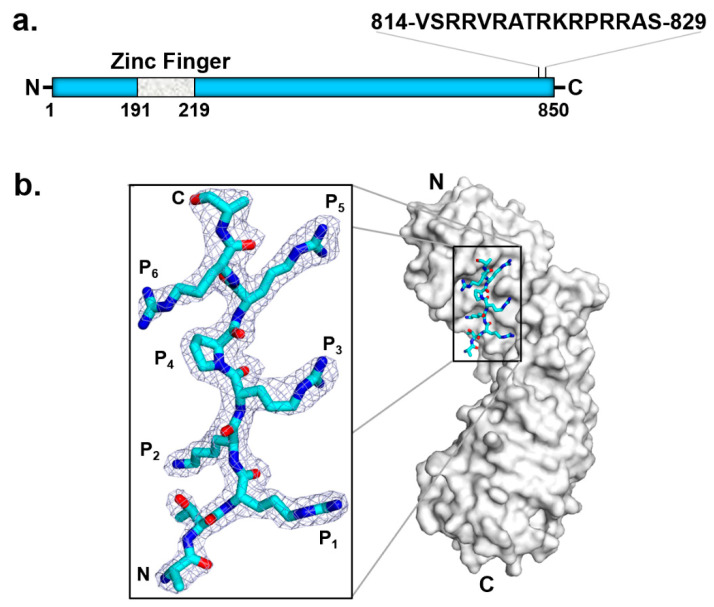
HCMV TerS has an NLS that binds importin α. (**a**) Schematic diagram of HCMV TerS highlighting conserved predicted motifs. (**b**) Crystal structure of importin α in complex with pUL56 NLS (zoom-in window) that binds the major NLS-binding pocket of importin α (PDB 5HUY).

**Table 1 viruses-14-02215-t001:** Inventory of 3D structures of 5TerS deposited in the Electron Microscopy Data Bank (EMDB) and PDB databases divided by virus family.

		Virus	Protomer M.W. (kDa)	Channel Diameter Min–Max (Å)	Number of Subunits	Accession Number(s)		Methodology		
						EMDB	PDB	NMR	X-ray	Cryo-EM
Bacteriophage	*Podoviridae*	P22	18.6	20–25	9		3P9A		+	
		Sf6	15.5	~19	8		4DYQ		+	
		PaP3	16.6	9–15	9		6W7T, 7JOQ		+	
	*Siphoviridae*	P74-26	18.7	~30	9	21,012	6V1I			+
		HK97	18.4	~18	9		6Z6E		+	
		G20C	18.8	~30	9		6EJQ, 4XVN		+	
		SF6 (SPP1-like)	16.0	11–29	9		3ZQM, 3ZQN, 3ZQO, 3ZQP, 3ZQQ, 4ZC3, 2CMP		+	
		Lambda	20.4	n/a	2 *		1J9I	+		
	*Myoviridae*	E217	21.3	22–52	10	26,858	7UXE			+
		44RR	17.3	24–32	11,12		3TXQ, 3TXS		+	
Herpesviruses		HSV-1	85.6	n/a	n/a	nd **	6M5S			+

* purified as a heterotrimeric 1TerL:2TerS complex. ** nd = not deposited. + = methodology used.

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
