# Peer review of "Viral Small Terminase: A Divergent Structural Framework for a Conserved Biological Function"

_viruses, 2022, doi:10.3390/v14102215_

Round 1

Reviewer 1 Report

The review deals with the interesting topic of the virus small terminase TerS, one of the three components of the DNA-packaging motor that is involved in the DNA packaging inside the capsid of bacteriophages and herpesviruses. This protein is the less known of the three subunits and several conundrum and questions remain about its essential properties and features. The involvement of TerS in the specific DNA binding and also in the stimulation of the ATPase activity of TerL and control of the nuclease activity of TerL have been demonstrated but the fine details are lacking. The review realizes a good job to expose these topics with an up-to-date report of the available structures and the clear identification of the unknown points.

Details :

In Figure 1a The legend is very technical and sounds like a 'Material and methods' section, while there is no a clear indication of the signification of the numbers shown, also the color signification (different families ?) is not explicit , similarly in Figure 1b,  while the indication of the signification for numbers are OK, the color code is not defined.

In the sentence at line 111 112 the authors mention that with a sequence identity of 16-20 % the herpesvirus TerS is a highly conserved protein, I thought, that with such values we are rather in the twilight zone (10-30%) where it is difficult to predict really similar structures and therefore is it really possible to talk about of a highly conserved protein ? But we recognize that it is clear that TerS from Herpesvirus is more conserved that TerS from Phages.

Line 246-248, We think that it could be useful for the reader that the differences (length, composition, etc ..) between the cos and pac sequences were clarified. Also, as the differences between the two packaging mechanisms (with cos and with pac sequences) are not well exposed, this section could be difficult to follow for a non-specialist reader.

In figure 3 . The legend did not seem to agree with the text lines 298-303, for PaP3 K19 mentioned in the text is not seen in figure (rather K18), also R49 for R48, R57 for K57, the reasons for these difference are not clear.

Author Response

REVIEWER #1

The review deals with the interesting topic of the virus small terminase TerS, one of the three components of the DNA-packaging motor that is involved in the DNA packaging inside the capsid of bacteriophages and herpesviruses. This protein is the less known of the three subunits and several conundrum and questions remain about its essential properties and features. The involvement of TerS in the specific DNA binding and also in the stimulation of the ATPase activity of TerL and control of the nuclease activity of TerL have been demonstrated but the fine details are lacking. The review realizes a good job to expose these topics with an up-to-date report of the available structures and the clear identification of the unknown points.

Details:

In Figure 1a The legend is very technical and sounds like a 'Material and methods' section, while there is no a clear indication of the signification of the numbers shown, also the color signification (different families ?) is not explicit, similarly in Figure 1b, while the indication of the signification for numbers are OK, the color code is not defined.

Agreed. We have updated the text on page 2, the Figure 1 legend, and removed colors from panel (b). The legend now reads

Figure 1. TerS conservation in bacteriophages and herpesviruses. (a) Phylogenetic trees of 20 representative phage TerSs (top panel) and eight pUL28-homologs (bottom panel). The scale bar represents the estimated evolutionary distance. The supporting numbers in the tree refer to the percentage of bootstrap distribution based on 1000 resamples. TerSs that share significant similarities are grouped and colored consistently. Seven groups of TerSs were identified among the 20 phages analyzed here, and only three for Herpesviruses. The sequences were aligned with Clustal Omega (53) and converted to FASTA/Phylip format using EMBOSS Seqret (54). The phylogenetic trees were analyzed in PhyML3.0 (55) and drawn using iTOL (56), which reroots at the mid-point. (b) Sequence identity and similarity between representative TerSs from the groups in panel (a) were calculated using SMS (57).

In the sentence at line 111 112 the authors mention that with a sequence identity of 16-20 % the herpesvirus TerS is a highly conserved protein, I thought, that with such values we are rather in the twilight zone (10-30%) where it is difficult to predict really similar structures and therefore is it really possible to talk about of a highly conserved protein? But we recognize that it is clear that TerS from Herpesvirus is more conserved that TerS from Phages.

16-20% sequence identify with high sequence similarity (>30-40%) and similar sequence coverage is not uncommon for highly conserved viral proteins. We edited the text to reflect the critique. Lines 110-112 now read: ‘TerS is an entirely different protein in herpesviruses, unrelated to phages TerS, and encoded by a gene about four times the size of phage TerS with no detectable sequence similarity (51, 52). Unlike phages, TerS is more conserved in herpesviruses (Figure 1a). Phylogenetic tree analysis of eight TerS sequences similar to Human Herpesvirus 1 (HHV-1, also known as Herpes Simplex Virus 1, or HSV-1) pUL28 readily sorted TerS into three subfamilies, corresponding to alpha-, beta-, and gamma-herpesviruses with sequence identity/similarity of 16-20% and 33-34%, respectively, and similar amino acid length (Figure 1b). In summary, TerS has diversified dramatically in bacteriophages, but it is more conserved in herpesviruses.

Line 246-248, We think that it could be useful for the reader that the differences (length, composition, etc ..) between the cos and pac sequences were clarified. Also, as the differences between the two packaging mechanisms (with cos and with pac sequences) are not well exposed, this section could be difficult to follow for a non-specialist reader.

We added the sequences of phage l cos end (5’-GGGCGGCGACCT-3’) and PaP3 cos DNA (5′-GCCGGCCCCTTTCCGCGTTA-3′) that binds PaP3 TerS with Kd ~10 mM. We also included the 22-mer pac sequence of phage P22 that has no detectable binding affinity for TerS in vitro.

In figure 3 . The legend did not seem to agree with the text lines 298-303, for PaP3 K19 mentioned in the text is not seen in figure (rather K18), also R49 for R48, R57 for K57, the reasons for these difference are not clear.

Thank you for pointing out the discrepancies. We have fixed them in the figure and text.

Reviewer 2 Report

The review by Lokareddy et. al on viral small terminase is a very valuable contribution to the field. TerS in bacteriophages is highly divergent and can exist in different oligomeric states while still performing the same function of DNA recognition. The manuscript describes the known TerS structures from bacteriophages well, discusses the conserved regions and how differences in the oligomerization helical core results in different oligomeric states. The difference in TerS role in cos (vs) pac packaging and the various TerS DNA interaction models are my favorites in the manuscript. Overall the manuscript is excellent and outstanding.

I have only a few minor comments.

1. The authors mention that TerS was fundamentally reinvented in Herpesvirus when compared to Bacteriophage TerS. What does reinvention mean ? Since there is no significant sequence similarity between them and they are different proteins of same function.

2. Line 150: Change turn-helix-turn to Helix-Turn-Helix

3. Will it be possible to include a figure for two ring pac synapses model of how TerS interacts with DNA

Author Response

The review by Lokareddy et. al on viral small terminase is a very valuable contribution to the field. TerS in bacteriophages is highly divergent and can exist in different oligomeric states while still performing the same function of DNA recognition. The manuscript describes the known TerS structures from bacteriophages well, discusses the conserved regions and how differences in the oligomerization helical core results in different oligomeric states. The difference in TerS role in cos (vs) pac packaging and the various TerS DNA interaction models are my favorites in the manuscript. Overall the manuscript is excellent and outstanding.

Thank you!

I have only a few minor comments.

  1. The authors mention that TerS was fundamentally reinvented in Herpesvirus when compared to Bacteriophage TerS. What does reinvention mean? Since there is no significant sequence similarity between them and they are different proteins of same function.

Agreed. We revised the text. Line 104 now reads: ‘TerS is a different protein in herpesviruses, unrelated to phages TerS, and encoded by a gene about four times the size of phage TerS with no detectable sequence similarity (51, 52).

Lines 110-112 now read: ‘In summary, TerS has diversified dramatically in bacteriophages, but it is more conserved in herpesviruses

  1. Line 150: Change turn-helix-turn to Helix-Turn-Helix

Done, thank you!

  1. Will it be possible to include a figure for two ring pac synapses model of how TerS interacts with DNA

Agreed. We now show the twin ring pac synapsis’ model in Figure 4c.